

# No anomalous supersaturation in ultracold cirrus laboratory experiments

Benjamin W. Clouser[1,2], Kara D. Lamb[1,3], Laszlo Sarkozy[2], Alexandra Nisenoff[2], Jan Habig[4], Volker Ebert[5], Harald Saathoff[4], Ottmar Möhler[4], and Elisabeth J. Moyer[2]

[1]Department of Physics, University of Chicago, Chicago, IL, USA
[2]Department of the Geophysical Sciences, University of Chicago, Chicago, IL, USA
[3]currently at Cooperative Institute for Research in the Environmental Sciences, Boulder, CO, USA
[4]Institute of Meteorology and Climate Research, Karlsruhe Institute of Technology, 76021 Karlsruhe, Germany
[5]Physikalisch-Technische Bundesanstalt, 38116 Braunschweig, Germany

**Correspondence:** Benjamin W. Clouser (bclouser@uchicago.edu); Elisabeth Moyer (moyer@uchicago.edu)

**Abstract.** High-altitude cirrus clouds are climatically important: their formation freeze-dries air ascending to the stratosphere to its final value, and their radiative impact is disproportionately large. However, their formation and growth are not fully understood, and multiple *in-situ* aircraft campaigns have observed frequent and persistent apparent water vapor supersaturations of 5%-25% in ultracold cirrus (T < 205 K), even in the presence of ice particles. A variety of explanations for these observations

have been put forth, including that ultracold cirrus are dominated by metastable ice whose vapor pressure exceeds that of hexagonal ice. The 2013 IsoCloud campaign at the AIDA cloud and aerosol chamber allowed explicit testing of cirrus formation dynamics at these low temperatures. A series of 28 experiments allow robust estimation of the saturation vapor pressure over ice for temperatures between 189 and 235 K, with a variety of ice nucleating particles. Experiments are rapid enough (~10 minutes) to allow detection of any metastable ice that may form, as the timescale for annealing to hexagonal ice is hours or

longer over the whole experimental temperature range. We show that in all experiments, saturation vapor pressures are fully consistent with expected values for hexagonal ice and inconsistent with the highest values postulated for metastable ice, with no temperature-dependent deviations from expected saturation vapor pressure. If metastable ice forms in ultracold cirrus clouds, it appears to lack the defects and interfaces that are assumed to produce differences in vapor pressure from hexagonal ice.

## 1 Introduction

As air rises into the stratosphere, it is freeze-dried by condensation as it passes through the coldest regions of the upper troposphere and lower stratosphere (UT/LS). The temperature-dependent saturation vapor pressure over ice therefore plays a strong role in setting the water vapor concentration of the stratosphere as a whole (e.g., Brewer, 1949), and in determining the abundance and characteristics of radiatively important tropical cold cirrus. Inadequate understanding of saturation vapor pressure, or incomplete relaxation of air to saturation, would result in excess stratospheric water and errors in both chemistry

models and radiative forcing calculations. For example, an apparent supersaturation of 20% at 190 K over expected values (from the Murphy-Koop parametrization, henceforth MK) corresponds to a difference of about 0.7 ppm $H_2O$. If uniformly



distributed, this additional stratospheric water would increase global surface radiative forcing by about 0.2 W m$^{-2}$ (Forster and Shine, 1999). Incomplete dehydration would also change the radiative effect of the cirrus produced by freeze-drying ascending air, but the magnitude and even sign of the effect are not well known. Reduced cirrus ice content would reduce longwave and shortwave cloud forcing, with opposing cooling and warming effects. Modeling studies show effects that are of

comparable magnitude to the direct effect of water but disagree on the sign (Gettelman and Kinnison, 2007; Tan et al., 2016). Furthermore, some explanations for observed supersaturations invoke novel forms of ice, that may have intrinsically different radiative properties than those of hexagonal ice (Murray et al., 2015). It is therefore important to understand the physics of ice nucleation and growth at the cold temperatures found in this region.

High apparent supersaturations within ice clouds have been measured in several in-situ campaigns in the UT/LS, most

frequently in the coldest ice clouds, with temperatures at or below 205 K (Krämer et al., 2009). From observations in the 2002 CRYSTAL-FACE campaign, Gao et al. (2004) reported supersaturations of 13% above 202 K and 35% below 202 K in cold cirrus and persistent contrails. Using water and particle measurements from the 2006 CR-AVE campaign, Lawson et al. (2008) found supersaturations in excess of 50% in cirrus clouds. Inai et al. (2012) found frequent supersaturations above 25% in cirrus clouds near the cold point using CALIOP data (Winker et al., 2007) and balloon-borne chilled-mirror hygrometer

measurements taken during the 2007 and 2008 SOWER campaign (Fujiwara et al., 2010). Some degree of supersaturation has also been observed in cirrus at warmer temperatures. Petzold et al. (2017) show that hygrometer data (Neis et al., 2015) from IAGOS-CORE, a campaign using instrumented commercial aircraft reaching minimum temperatures of 205 K, exhibit most probable values of supersaturation over ice within cirrus of 5-10%. A laboratory experiment in the AIDA (Aerosol Interaction and Dynamics in the Atmosphere) cloud chamber spanning a wide temperature range (243 K – 185 K) showed values close to

saturation, but with a systematic increase of ~6% with decreasing temperatures (Fahey et al., 2014).

Numerous explanations have been proposed for these observations. Many studies interpret them as true 'anomalous super-saturation', i.e., resulting from errors in our understanding of saturation vapor pressure and impossible to explain with standard microphysics. Explanations involving anomalous supersaturation include organic coatings on ice crystals (Cziczo et al., 2004a, b), glassy states (Zobrist et al., 2008; Kärcher and Haag, 2004), surface uptake interference due to ice binding with HNO$_3$ (Gao

et al., 2004, 2016), temperature- and supersaturation-dependent accommodation coefficients (Zhang and Harrington, 2015), and metastable forms of ice (Peter et al., 2006). Other studies suggest that no anomaly is necessary, and that measured super-saturations result only from dynamics, i.e. they occur when uptake rates on ice crystals are slow enough that the timescales of relaxation to saturation are long. Long timescales to achieve saturation may result from low particle numbers and small particle sizes found at low temperatures (Krämer et al., 2009; Rollins et al., 2016), or strong updrafts that lead to relaxation only to a

pseudo-equilibrium value (Petzold et al., 2017). The ATTREX campaigns of 2013-14 provided examples of apparent supersat-urations due to low particle numbers: supersaturations of up to 70% were observed in low-concentration cirrus ($< 100\ L^{-1}$), but not in those with high concentrations (up to $10000\ L^{-1}$), even at cold temperatures (190 K) (Jensen et al., 2013). These measurements demonstrate that saturations consistent with MK are at least possible in cold cirrus. Finally, instrumental error could explain all or part of the observed anomalies (Fahey et al., 2014).



Metastable ices with non-hexagonal crystal structure and elevated saturation vapor pressures could provide an explanation that encompasses the diverse body of field measurements. Laboratory measurements have identified metastable ices with saturation vapor pressures as much as 10.5% higher than hexagonal ice ($I_h$) at temperatures below 200 K (Shilling et al., 2006). The properties of metastable ice are, however, determined by its crystal structure, which can take different forms that may have

different vapor pressures. In the conditions found in Earth's atmosphere, ice forms layers of puckered hexagonal rings referred to as Ice I (Hobbs, 1974), which can be stacked in different ways: as mirror images of each other ($I_h$), shifted by half the ring width (cubic ice, $I_c$), or in a combination of both stacking sequences (stacking-disordered ice, $I_{sd}$) (Malkin et al., 2012). Note that much of the literature on cubic ice is now thought to have been measuring $I_{sd}$ (Malkin et al., 2015). Vapor pressure over metastable ice is poorly understood, and some modeling studies suggest that it depends less on the crystal's cubicity (fraction

of cubic stacking sequences) than on the number and type of imperfections within the crystal (Hudait et al., 2016; Lupi et al., 2017).

Laboratory measurements and computer simulations suggest that stacking disordered ice could form in the UT/LS, which experiences the coldest temperatures found in Earth's atmosphere. Measurements by multiple groups have found $I_{sd}$ forming in supercooled droplets, by both homogeneous and heterogeneous nucleation. Homogeneous nucleation of $I_{sd}$ was seen by

Murray et al. (2005), Murray and Bertram (2006), Murray (2008), and Malkin et al. (2012) in micron-sized water and solution droplets suspended in oil at temperatures of 170-240 K, and by Amaya et al. (2017) in nanodrops frozen during expansion of $N_2$ carrier gas. Malkin et al. (2015) observed heterogeneous nucleation of $I_{sd}$ in water containing solid inclusions, and Kuhs et al. (2012) reported that pure hexagonal ice formation was never observed below 190 K. The cubicity in laboratory-generated $I_{sd}$ samples is variable and depends on factors such as the freezing temperature, droplet size, and aerosol type and content, but

can be as high as 75% in atmospherically relevant temperature ranges. Simulations agree that ice frozen at 180 K should form $I_{sd}$, with Moore and Molinero (2011) producing two cubic ice layers for each hexagonal ice layer, i.e. a cubicity of 67%.

No experimental studies to date have measured the resulting influence of $I_{sd}$ on the saturation vapor pressure expected in cirrus clouds. Many of the $I_{sd}$ nucleation experiments provide no means of measuring the vapor pressure over ice (i.e. those experiments involving droplets suspended in oil). Nachbar et al. (2018) show that ice formed by crystallization from amorphous

solid water shows a significantly higher vapor pressure than MK at temperatures below 190 K, but it is unclear if this is $I_{sd}$. Studies that report the free energy difference between metastable and hexagonal ices vary widely in their estimates, likely because the types of imperfections and defects that affect vapor pressure are strongly influenced by experimental conditions. No experiments have addressed the ice that is subsequently grown through vapor deposition onto $I_{sd}$ crystals. Observations of cirrus clouds are inevitably made after at least some growth has occurred, and no existing experiments address how these crystals

behave as new ice layers are added. Recent modeling work on depositional ice growth at UT/LS temperatures suggests that ice should grow exclusively hexagonal layers, regardless of nucleation method, as long as supersaturation levels are moderate and temperatures are above 200 K (Hudait and Molinero, 2016), while below 200 K some stacking disorder can occur. The properties of metastable ices nucleated and grown in real atmospheric conditions remain only poorly understood.

Any metastable ice formed in the cold UT/LS region should persist long enough to be relevant for cirrus microphysics. Ob-

served transformation times for metastable ice into $I_h$ depend strongly on the surface area of the samples (Murray et al., 2005),





but for low-surface-area samples such as frozen droplets, the time can be quite long. Mayer and Hallbrucker (1987), Murray and Bertram (2006), and Kuhs et al. (2012) report annealing times of tens of minutes to hours over the UT/LS temperature range, and observe that by the termination of their experiments the transformation to $I_h$ is often still not complete, especially at lower temperatures. Observations of secondary indicators like crystal habit suggest that metastable forms of ice may nucleate

and persist for some time in the coldest parts of Earth's atmosphere, such as the UT/LS region (Murray and Bertram, 2006) and the polar stratosphere (Lowe and MacKenzie, 2008). Under these conditions, it appears $I_{sd}$ should form crystals with three-fold symmetry (Hallett et al., 2002; Kobayashi et al., 1987; Furukawa, 1982), and crystals with trigonal symmetry have occasionally been observed in Earth's atmosphere (Murray et al., 2015).

The IsoCloud experimental campaign allows us to characterize saturation vapor pressure during post-nucleation growth of

ice crystals in conditions characteristic of the UT/LS. The campaign consists of a series of cooling experiments with sample temperatures from 185-235 K, homogeneous nucleation of sulfuric acid (SA) and secondary organic (SOA) aerosols, and heterogenous nucleation on Arizona test dust (ATD), with number densities high enough to ensure that vapor can reach an equilibrium over the duration of each experiment. To reconstruct the saturation vapor pressure over ice, we use a box model and the observed properties of the ice cloud and chamber gas as cirrus grow and dissipate.

## 15   2   Methods

The 2012-2013 IsoCloud campaigns at the AIDA (Aerosol Interaction and Dynamics in the Atmosphere) cloud chamber involve a series of cirrus formation experiments designed to probe anomalous supersaturation. The chamber's pressure and temperature can be varied to replicate conditions throughout the UT/LS, and rapid pumping on the chamber simulates updrafts and can initiate nucleation. The chamber can be seeded with a variety of liquid aerosols and ice nucleating particles, and

houses a variety of instruments that make useful measurements supporting the study of gas-phase water vapor, such as ice particle number and total water concentration. An in depth discussion of the experiments, methods, and instruments used in the campaign can be found in Fahey et al. (2014) and Lamb et al. (2017). In the IsoCloud campaigns, 28 pseudo-adiabatic expansion experiments at temperatures between 185 and 235 K and pressures between 300 and 170 hPa were suitable for analysis. In these experiments, the chamber was seeded with Arizona test dust (ATD), sulfuric acid (SA), and secondary

organic aerosols (SOA) which allowed the study of both heterogeneous and homogeneous nucleation. We use measurements of water vapor and total water (vapor + ice) and an ice growth model to estimate the saturation vapor pressure over ice ($e_i$) for clouds from 185 to 235 K, with vapor pressure measurements in the coldest temperature regime (< 205 K) provided by the new Chicago Water Isotope Spectrometer (ChiWIS). The remainder of this section is divided into subsections discussing the characteristics of the instruments used in the analysis, the experiments included and the criterion for their inclusion, and the

model used to retrieve the saturation vapor pressure.



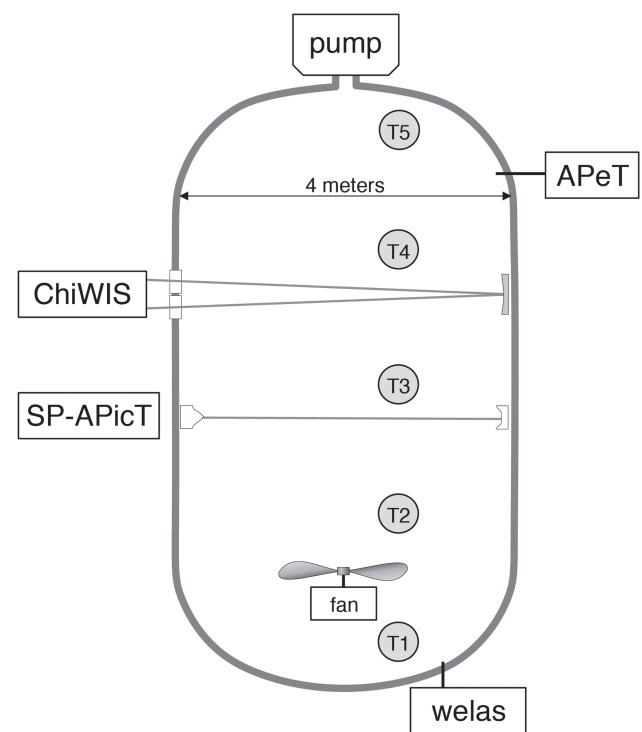

**Figure 1.** Layout of the instruments used in this analysis during the IsoCloud campaigns at the AIDA chamber. ChiWIS and SP-APicT, both open path tunable diode laser absorption spectroscopy (TDLAS) instruments, provided water vapor measurements. APeT, an extractive TDLAS instrument with a heated inlet, provided total water (ice + vapor) measurements, and Welas provided ice particle concentrations. The difference between total water and water vapor measurements was used to calculate ice mass in the chamber. Gas temperature is taken as the average of thermocouples 1 through 4. The whole chamber is within a thermally controlled housing that sets the base temperature of an experiment. The pumps draw gas out of the chamber in a pseudo-adiabatic expansion.

## 2.1 Instruments

Determining the saturation vapor pressure over ice requires measurements from three water instruments, an optical particle counter, and temperature and pressure sensors (Figure 1). Each of these measurements is described in the following sections, along with the instruments making them, typical accuracies and precisions, and limitations. Instrumental uncertainties are used to generate bounds on the retrieved saturation vapor pressures.

Our primary source of information is ChiWIS, a mid-infrared tunable diode laser instrument operated in open path mode using one of AIDA's White cell mirror systems. See Lamb et al. (2017) and Sarkozy et al. (in prep) for instrument details. The instrument has a typical precision of 22 ppbv in $H_2O$ in analyzed IsoCloud experiments for 1 Hz measurements at 299.2 hPa and 204.2 K, corresponding to relative precisions of 5% and 0.02% at 0.45 and 100 ppm, respectively. Measured quantities are retrieved by fitting spectra calculated from line parameters in the HITRAN database (Rothman et al., 2013) to raw





spectra, rather than by empirical calibration. Fitting is done using ICOSfit, a non-linear, least-squares fitting algorithm. Reported linestrength uncertainty of $\pm 5\%$ for the (413–524) line at 3789.63 cm$^{-1}$ in the $\nu_1$ band from which $H_2O$ measurements are derived contributes an additional $\pm 5\%$ systematic uncertainty in water measurements that is uniform across experiments. (Uncertainty due to temperature dependence is small in these experiments; see Section 3.5.)

The SP-APicT (single-pass AIDA PCI in cloud TDL) (Skrotzki, 2012) water vapor instrument is used to provide water vapor measurements in the case of thick ice clouds, which form in some IsoCloud experiments above 210 K (13 of 28 experiments). During warmer experiments which form very dense ice clouds, ChiWIS simultaneously experiences signal attenuation of up to 95% and backscattering of light off the cloud into the detector, producing artifacts that affect retrieved concentrations. During these intervals, we rely on the SP-APicT instrument to provide water vapor measurements because that instrument's single-
pass optical arrangement is much less sensitive to backscatter. At temperatures above 200 K, SP-APicT reports mixing ratios during ice-free periods about 2.5% lower than ChiWIS. For consistency across all experiments, we arbitrarily scale up the substituted SP-APicT measurements by that factor. (Details can be found in Lamb et al. (2017).) The resulting composite water vapor record uses ChiWIS measurements for 15 of 28 experiments, and scaled SP-APicT measurements for the remaining 13 experiments.

Total water measurements are provided by APeT (AIDA PCI extractive TDL), an extractive, tunable diode laser instrument (Ebert et al., 2008). In previous comparisons of AIDA instruments (Skrotzki, 2012), APeT measurements were found to be delayed by 17 seconds with respect to open-path *in-situ* TDLAS ones. As is standard practice, we take the chamber ice content to be the difference between total water and vapor phase measurements. To take advantage of the high precision ChiWIS affords at temperatures below 200 K, we use the composite water vapor record described above to calculate ice mass. However,
ChiWIS and APeT use different spectral lines and measurement methods, so care must be taken to ensure that ice mass is calculated correctly. Both instruments report values 2.5% below ChiWIS during ice free periods, so we scale up the APeT measurements by 2.5% as well. After applying this time-invariant scaling factor, if there is still an offset between the water vapor record and APeT total water prior to the expansion (when there should be no ice cloud in the chamber), that offset is subtracted from the whole experiment. These offsets are most significant below 200 K where they are typically between +0.05
and +0.25 ppm. Potential causes could include parasitic water absorption inside the instrument or outgassing from ice in the inlet of APeT (e.g. Buchholz and Ebert (2014)). See Supplemental Materials for details of instrument comparisons, and Table S1 for instrument offsets prior to pumping.

Ice particle concentrations are measured by the Welas 1 instrument. Ice particle number concentration is used to estimate the average radius of particles in the chamber and the average, per-particle growth rate. One component of this instrument's
uncertainty comes from $\sqrt{n}$ counting errors. However, experiments included in this analysis have high ice particle densities and small counting errors. We therefore neglect the counting errors in this analysis. The conversion of this instrument's count rate into a number concentration has a 10% uncertainty. In experiments where particles are very small, the Welas 1 instrument likely undercounts them since its efficiency drops sharply for particles below 0.7 μm in diameter (Wagner and Möhler, 2013). We address the steps taken to characterize that undercount in the following sections.



We assume a single chamber temperature at each point in time, and construct a value from the average of four thermocouples suspended at different heights in the chamber. These measurements have an apparent precision of 0.3 K during pumpdowns and 0.15 K during static conditions between pumpdowns (Möhler et al., 2003). A mixing fan at the bottom of the chamber is always operational and enhances the uniformity of the chamber, with a mixing time constant of about 1 minute. Figure 1 shows

the positions of the instruments used during IsoCloud in the AIDA chamber, as well as the locations of the thermocouples.

## 2.2   Experiments

IsoCloud expansion experiments were designed to nucleate, grow, and maintain cold cirrus clouds with the goal of testing for the presence of anomalous supersaturation under conditions similar to the coldest parts of the atmosphere. To be a suitable test for anomalous supersaturation, an expansion experiment must satisfy two basic criteria. First, its duration must be significantly

longer than the vapor relaxation times associated with cirrus growth. Relaxation times depend on experimental conditions, and are longer in cases where particle number densities are small and diffusion limitation is strong. Second, cirrus cloud growth must continue for long enough to allow for the retrieval of the saturation vapor pressure over ice. In practice, this means that the chamber's ice-covered walls must serve as a source of vapor from which the cirrus cloud can continue to grow throughout the experiment. The remainder of this section describes a typical expansion experiment, addresses the consequences of running

experiments in the presence of wall ice, and discusses the criteria for exclusion from analysis.

In a typical IsoCloud experiment, ice clouds are formed by pumping on a chamber filled with water vapor near saturation. Adiabatic expansion causes rapid cooling, which in turn leads to nucleation of ice. Air is kept close to saturation before pumping by preparing the walls with a thin coating of ice. In practice, chamber water vapor pressures are 80-90% of MK saturation before the expansions, which suggests that the wall ice is 0.5-2 K colder than the chamber air. Pumping and adiabatic expansion cool

the chamber air below the wall temperature, and given the presence of ice nucleating particles, the now-supersaturated chamber air will nucleate an ice cloud. Ice growth then draws the chamber vapor pressure below the saturation vapor pressure at wall temperature, and the walls become an additional source of water for the growing cirrus cloud. The transfer of mass from the walls is often large enough that chamber total water is greater at the end of pumping than at the beginning, despite loss to the pumps. Once the pumps cease, the chamber warms and the cirrus cloud dissipates and part of its mass is transferred through the

vapor back to the walls. The total amount of cooling in an experiment varies from 5-9 K, depending on pump speed, and occurs primarily during the first ~100 seconds of pumping when the chamber air behaves nearly adiabatically. Subsequently, heat flux from the walls becomes large enough to balance the adiabatic cooling. Cooling rates during the early stages of pumpdowns are equivalent to effective atmospheric updraft speeds of several meters per second, much faster than those typically associated with cold cirrus in the natural atmosphere.

Of the 48 IsoCloud experiments in March of 2013, six were reference expansions, and four others lacked measurements of one of the physical quantities required for analysis. Of the remaining 38 experiments, this analysis uses 28 and excludes 10. We include all experiments conducted with standard protocol in which the ice cloud can be reasonably expected to approach saturation. Five experiments are excluded for overly long relaxation times, and five for non-standard protocol. See Tables S1 and S2 for characteristics of included and excluded experiments, respectively. We estimate vapor relaxation times for each



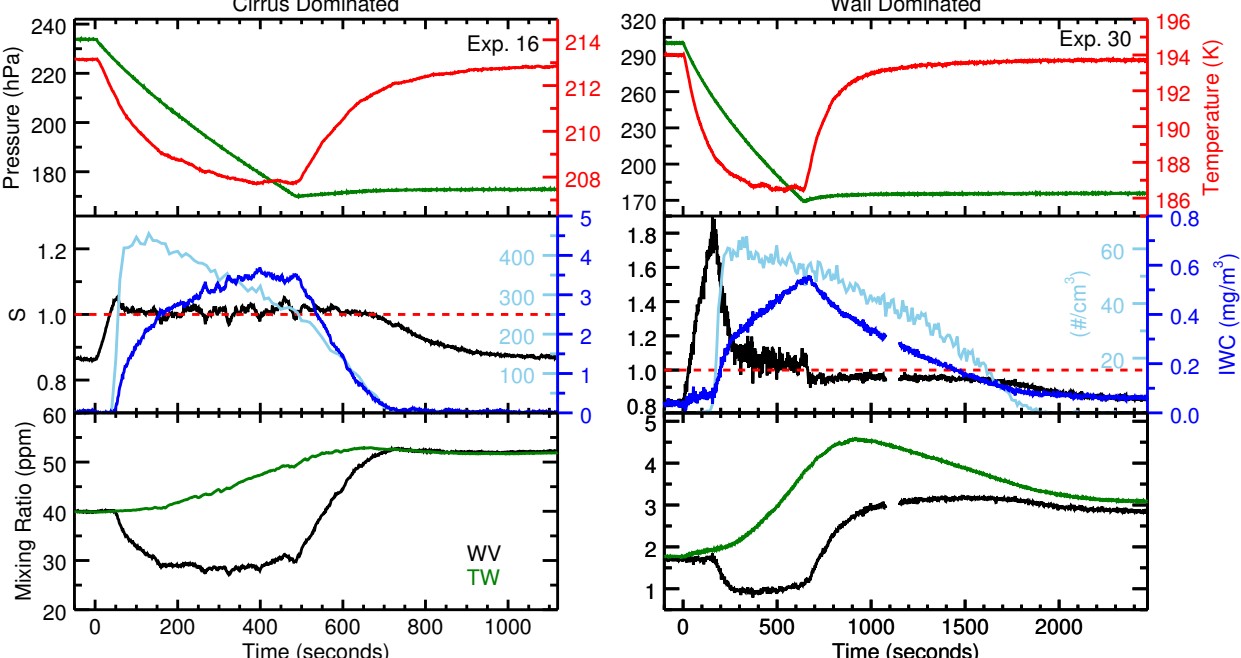

**Figure 2.** Examples of experiments in which vapor is controlled by cirrus uptake (*left*) and wall flux (*right*). Experiment 16 (*left*) is a heterogeneous nucleation experiment onto ATD, and Experiment 30 (*right*) is a homogeneous nucleation experiment with SA aerosol. The top panels of each plot show the pressure (green) and temperature (red) evolution. The action of the chamber pumps results in a pseudo-adiabatic expansion which cools the chamber gas rapidly at first, then more slowly until the cooling is balanced by heat flux from the chamber walls. The chamber gas warms as soon as the pumps have stopped. The middle panels show Murphy-Koop (MK) saturation (red, dashed line), measured saturation (black), ice particle number (light blue), and the cloud's ice water content (dark blue). Ice nucleation starts at the peak in saturation, and is followed by a sharp increase in particle number and rapid cloud growth. Saturation relaxes back to a constant value, where it stays until the pumps turn off. In the cirrus-dominated experiment, that value is the saturation vapor pressure over ice. In the wall flux dominated experiment, that value of about 8% supersaturation is that which is required to drive enough ice growth to balance the wall outgassing. When the pumps stop, the vapor pressure returns to the wall controlled value in the cirrus controlled experiment, but in the wall flux controlled experiment the decay rate is limited by wall uptake. The bottom panels show total water (green) and water vapor (black). The small data gap in Experiment 30 at around 1100 seconds is due to realignment of the chamber's White Cell mirrors.

point in each experiment using the expression of Korolev and Mazin (2003), which takes into account cooling rate (effective updraft speed), ice particle number, and particle size to estimate the timescale for achieving a pseudo-equilibrium value. (See Section S2.3 for expression.) For each experiment, we determine $\tau_{min}$, the minimum relaxation time at any point during the experiment, and $t_{exp}$, the time interval over which the calculated relaxation time is within a factor of two of $\tau_{min}$. We consider

5  that an experiment should reasonably approach pseudo-equilibrium if $t_{exp}/\tau_{min} > 4$.

The five non-standard experiments excluded are Experiment 1, which had an abnormally short pumping time, and Experiments 40-43, where the chamber was prepared with dry walls. Pumping in Experiment 1 lasted only 250 seconds, vs. 400-750





seconds in all other experiments; we would expect more inhomogeneities in the resulting ice cloud. Experiments with dry walls pose a problem for our analysis because the lack of an ice source means that these experiments do not involve extended periods of ice growth near saturation.

The 28 experiments used in this analysis still show a range of characteristics, and can be grouped into two broad categories. In 'cirrus-dominated' experiments (see Figure 2, left, for example), the wall flux is comparable to ice uptake driven simply by the change in saturation vapor pressure on cooling. In these experiments water vapor concentrations draw down quickly to saturation. In the colder experiments, however, wall flux is generally far more substantial. In these 'wall-dominated' experiments (Figure 2, right), peak total water rises to many times greater than initial water vapor, water vapor remains supersaturated during the growth phase of the experiment, and then becomes subsaturated during evaporation. This deviation complicates analysis and requires a growth model to determine saturation vapor pressure. For consistency, we treat all experiments the same, and extract saturation vapor pressure using the same method.

The two examples shown in Figure 2 illustrate the key features of each type of experiment. In the 'cirrus-dominated' experiment (Figure 2, left), the onset of nucleation produces rapid ice growth and a corresponding drawdown of vapor pressure to a value close to saturation. The ice cloud then grows slowly for the remainder of the expansion experiment, with water provided by the ice-covered chamber walls. In this particular case, a heterogeneous nucleation experiment with abundant ice nucleating particles, ice nucleation occurs at a relatively low supersaturation, and the ice particle number reaches ~400 cm$^{-3}$ before decreasing nearly in proportion to the action of the pump. After the pumping stops, the ice cloud decays over a roughly 200 second period, and chamber vapor pressure returns to the wall-controlled value.

In the 'wall-dominated' experiment (Figure 2, right), significant supersaturation with respect to MK persists throughout the experiment. In this particular case, initial supersaturation is quite high since the chamber was prepared with only sulfuric acid droplets to study homogeneous nucleation. The onset of nucleation again produces a drawdown of supersaturation, but only to a value of about 8%, which remains fairly constant for the duration of pumping. This is the value required to drive the strong continuing ice growth that balances the wall flux. Once the expansion stops, the chamber air warms, the walls become a water sink rather than a source, and chamber vapor pressure drops to RH$_{ice}$~95%, the value required to drive enough evaporation to balance wall uptake. After the ice cloud has nearly dissipated, the chamber vapor pressure again returns to the wall-controlled value.

These chamber dynamics mean that saturation vapor pressure in IsoCloud experiments cannot be determined simply by measuring the water vapor content in the chamber after an ice cloud has developed. The colder the experiment, the more wall-dominated it typically becomes, so that experiments show steadily increasing long-term supersaturations with respect to Murphy-Koop saturation (MK) as temperature decreases, rising by approximately 5% over the temperature range 225-185 K (Figure 3). This rise should not be interpreted as the result of a temperature-dependent saturation vapor pressure, but instead as a temperature-dependent balance between wall flux and diffusional ice growth. The wall flux contribution is relatively larger for colder experiments because saturation vapor pressure falls sharply with reduced temperature and the diffusional ice growth rate drops with vapor concentration. The result is that the colder the temperature, the larger the supersaturation over ice required to produce steady-state relative humidity. For this reason we use an ice growth model to retrieve saturation vapor




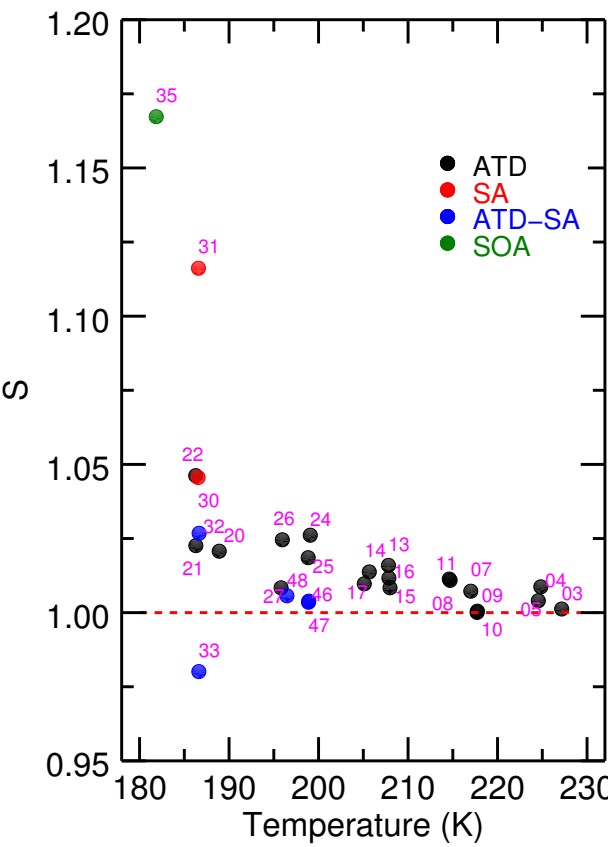

**Figure 3.** Average measured saturation of the 28 IsoCloud experiments after relaxation back to a near-constant value. Supersaturations plotted are the average value of the final 200 seconds of pumping. Experiments are colored by aerosol/IN type: Arizona test dust (ATD, black), liquid sulfuric acid droplets (SA, red), secondary organic aerosol (SOA, green), and experiments containing both ATD and SA (blue). Warm, 'cirrus-dominated' experiments ($T \geq 195$ K) typically show vapor pressures close to MK, with saturations from 1.00 to 1.02. Cold, 'wall-dominated' experiments (below ~195 K) show saturations that rise with decreasing temperature. Higher supersaturation is necessary for the cirrus growth rate to match the mass flux off the chamber walls. This effect means that an ice growth model is necessary to extract the saturation vapor pressure. Water vapor retrievals during the pumping interval in Experiment 33 are subsaturated with respect to MK.

pressure by modeling vapor pressure evolution during each experiment. The model, fitting procedure, and uncertainty analysis are described in Section 3.

## 3  Analysis

We model the vapor pressure evolution during each experiment assuming diffusional growth to a sphere (Equation 1). Model

5  inputs are all measured or derived from measured quantities — ice mass, particle number, growth rate, pressure, and temper-





ature — with saturation vapor pressure as the single free parameter. That is, we assume saturation vapor pressure over ice is $e_{sat} = x e_i$, where $e_i$ is the Murphy-Koop saturation vapor pressure (Murphy and Koop, 2005) and $x$ is a constant scale factor separately fit for each experiment. The model predicts the evolving chamber vapor pressure, and we fit that prediction to the observed $H_2O$ vapor pressure, minimizing the difference between observed and calculated values.

## 3.1 Ice growth model

The model is obtained by rearranging an expression for the diffusional growth rate over ice (Pruppacher and Klett, 1997) to calculate the far-field water vapor pressure:

$$e = x e_i \left(1 + \frac{\dot{m} L_i}{4\pi \bar{r} k_a^* T_\infty} \left(\frac{L_i M_w}{R T_\infty} - 1\right)\right) + \frac{\dot{m} R T_\infty}{4\pi \bar{r} \alpha D_v^* M_w} \tag{1}$$

Measured and derived quantities here are $\dot{m}$, the per-particle growth rate (change in total ice mass/time/particle #); $\bar{r}$, the average particle radius; $T_\infty$, the gas temperature in the chamber; and we identify the far-field vapor pressure $e$ as the measured vapor pressure. Parameters are $M_w$, the molar mass of water; $L_i$, the latent heat of sublimation; $D_v^*$, the diffusivity of water in air with kinetic corrections; $\alpha$, the accommodation coefficient; and $k_a^*$, the thermal accommodation coefficient, which is taken here to be unity (Fung and Tang, 1988). The average radius of the ice particles, $\bar{r}$, is calculated from the total ice water mass and particle number counts described previously, and a temperature-dependent ice density. The bulk density of ice varies by about 1% between -10 and -100 C; the values used in this work are from a quadratic fit to data from Eisenberg et al. (2005), which are based on the x-ray diffraction measurements of La Placa and Post (1960). We assume the particles are spherical, which is a reasonable approximation for small, micron-sized particles. The diffusivity of water vapor in air, $D_v^*$, is also temperature dependent, and is evaluated using the functional form of Pruppacher and Klett (1997), which includes kinetic corrections (Okuyama and Zung, 1967; Fitzgerald, 1972). Note that one limitation of this method is that it can yield only a bulk value, and is not sensitive to situations in which a small subset of ice crystals are metastable.

## 3.2 Confounding issues and corrections

We apply sensitivity tests or corrections to three issues that might confound analysis: loss of ice crystals by pumping, uncertainty in the accommodation coefficient, and undercounting of particles. The issues are sufficiently unproblematic that they are not included in the formal uncertainty analysis of Section 3.5.

The pseudo-adiabatic expansion procedure during experiments results in a loss of ice mass as air is removed from the chamber. This loss must be accounted for in order to accurately estimate ice mass change by sublimation/deposition to/from the vapor. The pumps remove a constant volume of gas from the chamber in each time interval, and we assume that they act in the same manner upon the small ice particles found in our experiments. With this assumption, we correct the ice growth rate by subtracting the assumed pumping losses from the derivative of the cirrus ice mass.

The accommodation coefficient $\alpha$ in Equation 1 is not well constrained, with significant variation in the literature. $\alpha$ can be thought of as the probability that a molecule of water vapor that strikes the surface of an ice particle is incorporated into the ice matrix, and can be sensitive to experimental conditions. For similar chamber experiments, studies have shown that





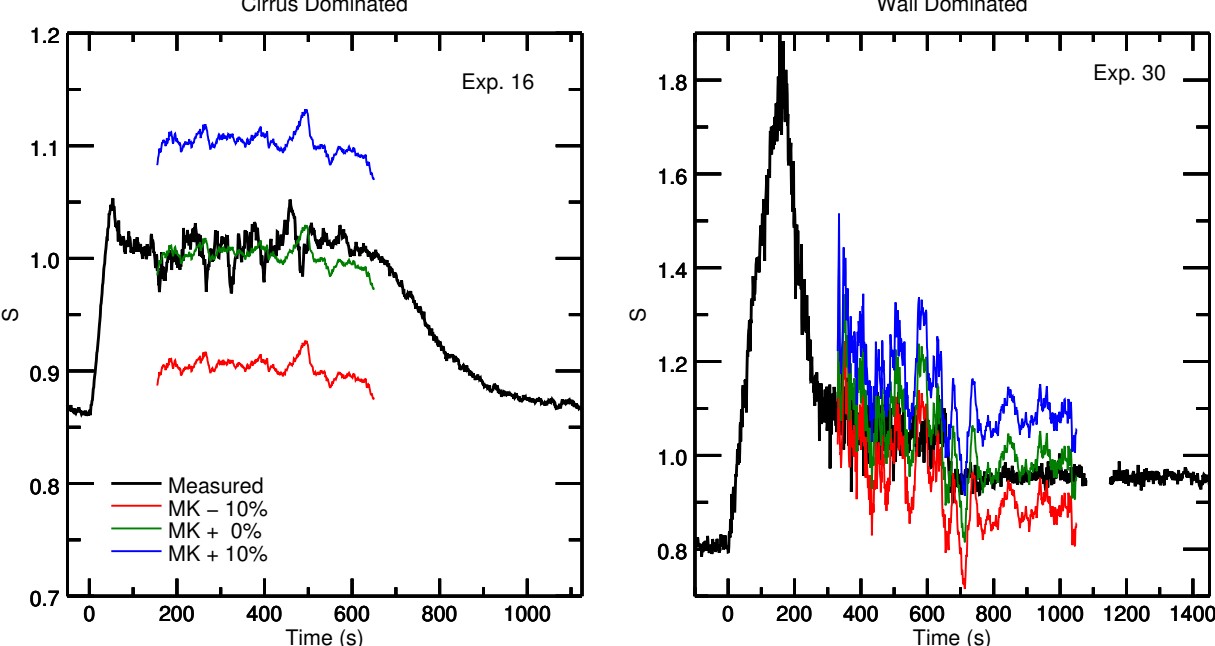

**Figure 4.** Model output for experiments used in the previous example, cirrus-dominated experiment 16 (*left*) and wall-dominated experiment 30 (*right*). Each output is calculated using observed quantities under three different assumptions about the 'true' saturation vapor pressure of cirrus. The red line assumes the true value is 10% lower than the MK saturation vapor pressure, the blue line assumes it is 10% higher, and the green line assumes MK is the true saturation vapor pressure. Calculated saturations are smoothed by 30 points (~30 seconds). Measured saturations are unsmoothed. Experiments are very sensitive to the assumed vapor pressure, although wall-controlled experiments are noisier overall since they are typically at lower temperatures. Experiment 16 shows spikes in measured water which are due to real sampling of different air masses during the turbulent period when the pumps are on and the cirrus cloud is growing. Experiment 30 shows oscillations in the model output due to in-mixing of warmer air from the top of the chamber. This type of temperature fluctuation is not captured by the model.

the accommodation coefficient can be treated as a constant during an experiment (Lamb et al., in prep) with values close to 1 (Skrotzki et al., 2013; Lamb et al., in prep). We test the sensitivity of our vapor pressure model to uncertainty in the accommodation coefficient by running the model with different values of $\alpha$, and find that the derived results for saturation vapor pressure are quite insensitive to the exact values of $\alpha$ in the range of 0.2 to 1 (Supplementary Materials Figure S8). We therefore use a value of 1 throughout this work, but note that if the true $\alpha$ value is below 0.2, then this assumption will result in an overestimate of the saturation vapor pressure.

Undercounting of particles may occur because the Welas 1 optical particle counter has a size cutoff for small particles of 0.7 µm in diameter. In these experiments, we never see evidence of very large particles, but very small particles are common at the beginnings and ends of experiments, immediately after nucleation or towards the end of sublimation, respectively. For some experiments at the coldest temperatures, where initial water vapor and final ice mass are small, we also expect undercounts





throughout the experiment. Mean particle size in these experiments is strongly temperature-dependent, ranging from ~1 micron at 189 K to ~5 microns at 235 K. Failure to account for undercounting would lead to an overly large average radius, and could produce a low bias in retrieved saturation vapor pressures.

We deal with the undercounts in two ways: we exclude all time periods in which the calculated average radius is less than 0.85 microns, and we conduct sensitivity tests on resulting analyses. In several of the colder experiments, however, the mean calculated radius remains below 1 micron throughout the experiments. In these marginal cases assuming a log-normal distribution produces estimated undercounts of up to 50% throughout the experiment (See Supplementary Material for details of this calculation). We therefore conduct sensitivity analyses on all experiments of uncertainty due to potential undercount by increasing ice particle counts by factors of 1.5, 2, and 5. (See Supplementary Material, Figure S9). Undercounting can result in underestimation of saturation vapor pressure, but most IsoCloud experiments show sensitivity of retrieved saturation vapor pressure of less than +0.5%, even in the unrealistic case of undercounting by a factor of 5. Maximum sensitivity to undercounting occurs in the three homogeneous nucleation experiments, where particle sizes are smallest, but still remains under +2.5% even in the most extreme case tested.

### 3.3 Ability to diagnose saturation vapor pressure

Before fitting our experimental data, we conduct a preliminary proof-of-concept exercise to evaluate whether the vapor pressure model is indeed sensitive to assumptions about saturation vapor pressure. We calculate evolution of the chamber vapor pressure during selected representative experiments under three different assumptions of saturation vapor pressure values: MK saturation, and MK multiplied by factors of 1.1 and 0.9. Comparing these calculations to the observed values, we see that even small changes in the assumed saturation vapor pressure result in significant deviations from the measured chamber water vapor in both cirrus-dominated and wall-dominated experiments (Figure 4). Results suggest that experiments are sufficiently sensitive to resolve differences in saturation vapor pressure of a few percent. This test establishes that observations of chamber vapor pressure during ice growth can in fact constrain the saturation vapor pressure in all the IsoCloud experiments.

### 3.4 Fitting procedure and region choice

The model is fit to the observed chamber vapor pressure using least-squares optimization. We use MPFIT (Markwardt, 2009), a Levenberg-Marquardt least-squares minimization routine written in IDL, based on the MINPACK algorithm (Moré, 1978). This routine attempts to minimize the difference between the model and observation by varying the scaling factor $x$ in Eq. 1, which multiplies the Murphy-Koop parametrization of the saturation vapor pressure over ice. The fit routine yields a single value of $x$ for each experiment, which is multiplied by MK saturation to best fit the observations.

The fit region for each experiment is selected using three criteria. 1) The fit region must start after the maximum ice particle number count has been attained. In most experiments, the maximum particle count is achieved within about 50 seconds of the peak in saturation associated with the onset of nucleation. During the preceding brief period of rapid ice growth, significant particle undercounts are likely. 2) We exclude all time periods when the Welas 1 instrument reports fewer than 12 ice particles per $cm^3$. This criterion typically excludes the late portions of experiments when the ice cloud has almost completely decayed.





3) We exclude all the time periods in which the average particle radius is less than 0.85 microns, as described in Subsection 3.2, again because particle undercounts are likely. This criterion becomes relevant for cold experiments, in which vapor pressures are low and particles grow slowly and remain small. (In IsoCloud experiments at temperatures below 195 K, average particle radius remains under 1.5 microns at all times.) These criteria result in an average fit region length of ~700 seconds. The shortest

fit region is 259 seconds (Experiment 9) and the longest fit region is 1647 seconds (Experiment 21). Colder experiments tend to have longer fit regions, since in these experiments the ice cloud can linger for tens of minutes after pumping has ceased.

### 3.5   Uncertainty Analysis

We calculate error bars for each experiment that reflect uncertainties from several sources: intrinsic measurement precision for water vapor and other key observables, uncertainty due to experimental artifacts (e.g. chamber inhomogeneities that affect

model fits), and systematic offsets (e.g. line strength errors that produce multiplicative errors in derived vapor pressures). We group the first two categories, measurement precision and chamber artifacts, under the term 'instrumental uncertainty'.

    Instrumental uncertainty for each experiment is calculated using a Monte-Carlo method: for each experiment, we generate 2000 parameter sets in which each physical parameter used in the calculation is randomly drawn from its estimated distribution, and then run the fit routine on each set. See Table S3 for the list of parameters varied and their assumed distributions. AIDA

chamber temperature uncertainty of 0.3 K is included in this table as a random variable, although the temporal correlation of temperature errors is not well-known. Resulting error bars may therefore be slightly underestimated. The resulting distribution of saturation vapor pressure values is nearly normal in each experiment, and we take its standard deviation to be the component of the error bar associated with instrumental uncertainty.

    The primary source of systematic offsets is uncertainty in the spectroscopic line strengths used to retrieve water vapor

concentrations from the observed spectral features. All spectroscopic parameters are taken from the HITRAN 2012 database (Rothman et al., 2013), which provides uncertainty estimates for each parameter. Line strength errors arise in two ways: through raw uncertainty of the measured line strength at the reference temperature of 296 K, and through uncertainty in the measured experimental gas temperature that propagates to uncertainty in the calculated temperature-dependent line strength. The ChiWIS instrument uses the $H_2O$ line at 3789.63 cm$^{-1}$, which has a stated $1\sigma$ uncertainty of $\pm5\%$ in HITRAN 2012. AIDA chamber

temperature uncertainty is assumed to be randomly distributed, but even if it were systematic it would contribute an additional line strength uncertainty of only 0.1% and so is not included in this analysis. (Note that typical temperature declines of 5-9 degrees during expansion experiments are automatically incorporated in the retrievals.) Raw line strength errors are added directly to the calculated error bars, and in all but the coldest experiments are the dominant source of uncertainty.

    Because the final output of the analysis is the relationship of saturation vapor pressure to temperature $x(T)$, we must consider

a final source of uncertainty, that each experiment produces a single value for $x$ but spans several degrees of cooling. We therefore construct horizontal error bars to acknowledge the spread in $T$, assigning them the standard deviation of chamber temperatures during the experimental fit period. These error bars are typically smaller in warmer experiments, since fit regions in that regime lie almost completely within the time interval when the wall heat flux balances the adiabatic cooling. Horizontal





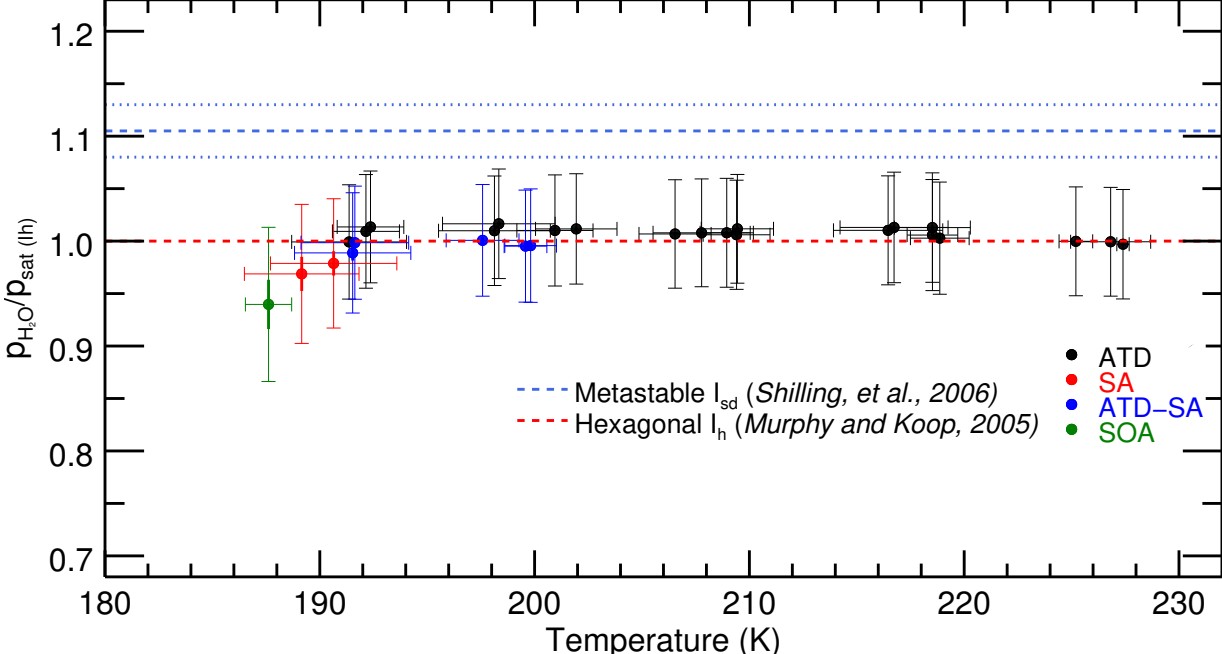

**Figure 5.** Retrieved saturation vapor pressures for the 28 IsoCloud experiments fitted, expressed as a fraction of MK saturation, and plotted against mean experiment temperature. The red, dashed line represents MK saturation. The blue, dashed line represents Shilling's value for the vapor pressure of metastable ice, and the blue, dotted lines show the errors of that measurement. Experiments are colored by aerosol/IN type: Arizona test dust (ATD, black), liquid sulfuric acid droplets (SA, red), secondary organic aerosol (SOA, green), and experiments containing both ATD and SA (blue). All experiments that include solid dust (black, blue) undergo heterogeneous nucleation; those with SA or SOA only (red, green) undergo homogeneous nucleation. Horizontal error bars are the standard deviation of the temperatures over the fit region. Vertical error bars show both the $1\sigma$ instrumental uncertainty (thick width), which is greater at colder temperatures, and the larger systematic linestrength uncertainty (thin width), which is identical for all experiments. Derived saturation vapor pressures in general are consistent with MK (to within systematic uncertainty), and exhibit no trend with temperature for each aerosol type (to within instrumental uncertainty). Note that the fitting procedure means that Experiment 33, the outlier in Figure 3, yields a saturation vapor pressure consistent with other experiments. Experiments do show lower values in cases where liquid aerosols are present. Compare to Figure A2 in Fahey et al. (2014).

error bars are larger in colder experiments since the ice cloud ofter persists for some time after the pumps turn off and the chamber begins to warm back to its base temperature.

## 4  Results

Results of fitting the IsoCloud experiments show a saturation vapor pressure consistent with MK, with no increase in retrieved
5  saturation vapor pressures at low temperatures (Figure 5). All experiments are inconsistent with the range of vapor pressures given for metastable ice by Shilling. Some differences are apparent between experiments with different ice nucleating particles,



so we focus first on those with only solid particles (Arizona test dust, black points in Figure 5). These experiments cover a temperature range from 235-193 K and have low $1\sigma$ instrumental uncertainty of less than +0.5%. (See figure S6 in the Supplementary Material, which shows model results labeled by experiment number and plotted with only instrumental uncertainties.) They show no temperature-dependent effects that could explain anomalous supersaturations observed in field experiments.

Derived saturation mixing ratios throughout the experimental temperature range are all consistent to within $2\sigma$ instrumental uncertainty (i.e. 1% of MK).

To test more carefully for any trend in saturation vapor pressure with temperature, we also perform a total least squares 2-parameter fit on the ATD model results. This line-fitting method takes into account uncertainty in both variables (in our case, experimental temperatures and instrumental uncertainties) rather than ascribing uncertainty only to a dependent variable. The fit

yields an intercept of $100.8\% \pm 0.1\%$ of MK at the mean temperature of 209.3 K, and a slope of $-0.04\% \, K^{-1} \pm 0.010\% \, K^{-1}$. The experiments are consistent with MK to well within their $\pm 5\%$ systematic uncertainty. The fitted trend with temperature is significant, but very small, equivalent to a change over the 40 degree IsoCloud range of $0.9\% \pm 0.4\%$ of MK saturation. It is driven largely by the grouping of points at 225-228 K; if these are disregarded, the fitted trend is no longer significant, only $-0.008\% \, K^{-1} \pm 0.01\% \, K^{-1}$.

The experiments performed at the highest temperatures in IsoCloud also demonstrate that the ice growth model used in this work does not introduce artifacts into the retrieved saturation vapor pressures. In these experiments (3-17, at $T = 205 - 235 \, K$), ice particle number is high and ice cloud growth dominates, so the chamber vapor pressure should draw down quickly to saturation. The values to which these experiments relax (shown in Figure 3) are effectively identical to those derived in our more complex analysis procedure: 100.0-102.0% in the simple calculation of Figure 3, and 99.7-101.2% in the fits of Figure 5.

This similarity confirms that the use of an ice growth model does not bias the derived saturation vapor pressure values.

Experiments in which liquid aerosols are present result in derived saturation vapor pressures lower than Murphy-Koop. The homogeneous nucleation experiments (red, green) show the strongest effect, but even those heterogeneous nucleation experiments with sulfuric acid aerosols present (blue) show lower vapor pressures than do experiments with only solid ice nucleating particles (black). To emphasize the difference between ATD experiments and experiments containing liquid aerosols,

Figure 6 shows just experiments below 205 K, with only instrumental uncertainties. ATD experiments are on average 0.8% above MK; ATD-SA points are on average 0.4% below; and the experiments with only liquid aerosols present — the three homogeneous nucleation experiments 30, 31, and 35 — show even lower fitted saturation vapor pressure, at 2-6% below MK. These experiments do have large uncertainties, as they are cold and dry, but their deviation from MK exceeds the $1\sigma$ instrumental error in all cases.

A total least squares fit to the ATD-SA experiments confirms that they are significantly different from ATD experiments without sulfuric acid aerosols present. ATD-SA experiments show no significant temperature dependence in deviation from MK saturation vapor pressure (fitted slope $= 0.01\% \, K^{-1} \pm 0.02\% \, K^{-1}$), but the fitted intercept yields an offset exceeding the $1\sigma$ instrumental error. The intercept at the ATD-SA mean temperature of 196.0 K is $99.7\% \pm 0.2\%$, which is significantly lower than the expected ATD values at that temperature of $101.3\% \pm 0.2\%$.





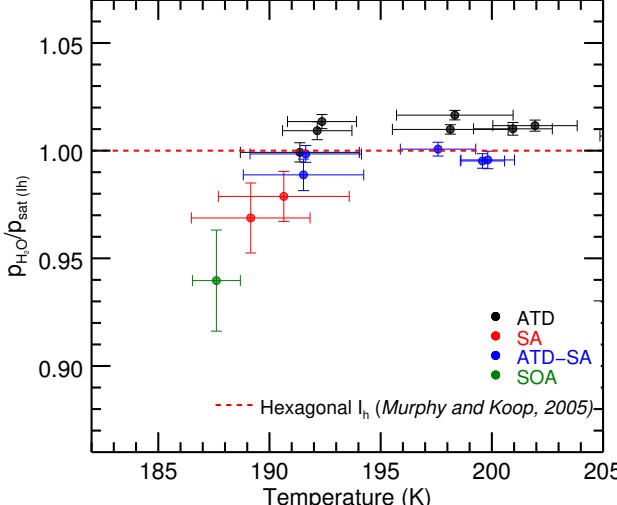

**Figure 6.** Zoomed in view of the experiments below 205 K. Linestrength errors result in the same shift for all experiments, so they are not included in the error bars here. Experiments containing sulfuric acid are ~2.6% lower on average than those containing pure ATD experiments. Experiments are colored by aerosol/IN type and MK saturation line is included for reference.

These results suggest that liquid aerosols introduce some additional factor that depresses implied saturation vapor pressure in our analyses. Although these experiments take place at cold temperatures and are probably subject to undercounting, the tests of sensitivity to ice particle number described in Section 3.2 indicate that undercounting cannot completely account for the observed differences. One possible mechanism may be water uptake by hygroscopic SA and SOA aerosols which remain
5 unfrozen during the experiments. Such a population of aerosols would have a vapor pressure lower than that of ice. This effect would not be captured by the ice growth model, and would systematically lower retrieved saturation vapor pressures.

## 5 Discussion and Conclusions

We see no evidence of the anomalous supersaturation in ultracold cirrus formation that field measurements had suggested. Our results show no temperature-dependent changes in retrieved saturation vapor pressure that could explain field observations.
10 Those colder experiments that show anomalies show low, rather than high saturation vapor pressures, which are likely artifacts resulting from the presence of liquid aerosols. For experiments with only solid ice nucleating particles, retrieved saturation vapor pressures are essentially identical throughout the 185-235 K experimental temperature range. Results are consistent with the MK parametrization throughout, with a mean value of MK + 0.8%, well within the 5% linestrength uncertainty. Scatter in experiments is small, and all experiments are inconsistent with the parametrization given by Shilling.
15 These results suggest that field measurements of anomalous supersaturation at low temperatures are most likely either the consequence of dynamical effects or of experimental error, or some combination of both. In heterogeneously nucleated cirrus with sparse nuclei, ice growth times may be so slow as to leave persistent observable supersaturation on the timescales of



natural temperature fluctuations. For example, Jensen et al. (2013) report *in-situ* observations during the ATTREX campaign of thin cirrus with low particle number densities (~0.01 cm$^{-3}$), supersaturations up to 70%, and estimated relaxation timescales of hours or longer. Krämer et al. (2009) summarize 20 high-altitude aircraft flights and report frequent supersaturation in cirrus, but also low number densities (~0.01 cm$^{-3}$) and estimated relaxation timescales of hours to days.[1] The possibility that

experimental error contributes to some observations of anomalous supersaturation cannot be entirely eliminated. Measurements of water vapor in the UT/LS are notoriously difficult due to the cold temperatures found there, and just 1 ppm of contaminating $H_2O$ in an instrument (due to inlet icing or outgassing) could lead to an 'anomalous supersaturation' signal of ~25% in a 190 K cold cirrus cloud at ~17 km in the TTL.

The IsoCloud experiments suggest that metastable forms of ice need not be considered in cloud models since they are

either not forming, or the types that are forming do not exhibit a vapor pressure significantly different from $I_h$. Many studies have suggested that metastable ice should nucleate and persist in cirrus at these temperatures. While Hudait and Molinero (2016) suggest from modeling studies that vapor-deposited ice should be hexagonal above 200 K, their work leaves open the possibility that metastable ice could form at colder temperatures. The experiments discussed here imply that if metastable ice does form, it must be free of the defects and imperfections that are assumed to result in higher vapor pressures than

hexagonal $I_h$ (Hudait et al., 2016; Lupi et al., 2017). The experiments shown here place strong constraints on ice formation in the atmosphere, because the rapid cooling times in IsoCloud should be maximally favorable to creating these defects, and the short experimental timescales mean that we should detect its effects before any annealing to hexagonal ice. These results suggest that even if metastable ice does form in the UT/LS region, its effect on vapor pressure and on transfer of water to the stratosphere would be negligible.

Although these results suggest that $I_{sd}$ cannot induce anomalous supersatuation in the UT/LS, $I_{sd}$ may nevertheless be of climatic importance if its radiative properties differ from those of hexagonal ice. Preliminary findings by Murray et al. (2015) suggest that trigonal crystals, which are associated with $I_{sd}$, have a lower absorption efficiency than hexagonal ones, and that for column crystals in particular over a broad range of sizes, trigonal column crystals have a significantly larger single-scattering albedo than do scalene column crystals or hexagonal column crystals. Since saturation vapor pressure seems not to provide an

indication of the presence of $I_{sd}$, further experiments would be needed to determine the conditions under which $I_{sd}$ may nucleate and grow under deposition in the ultracold regions of the UT/LS. High time-resolution diffraction measurements paired with observations of atmospherically relevant observables like water vapor pressure and crystal habit offer one possible method of probing the presence of $I_{sd}$. Moreover, the type of ice that first nucleates may influence crystal habit even if subsequent deposition is of purely hexagonal ice (Furukawa, 1982). Exploring the conditions in which $I_{sd}$ and metastable ices can form

in real atmospheric conditions may then also be important for understanding their radiative importance and possible future changes.

---

[1]Note that in this work, where we use particle number densities of order 50 cm$^{-3}$, our need for an ice growth model stems not from long relaxation times, but from the additional wall ice source not present in atmospheric cirrus.




*Data availability.* The IsoCloud datasets can be found at https://publish.globus.org/jspui/handle/11466/247.

*Author contributions.* B.W.C and E.J.M. led the data analysis; E.J.M. directed the construction and operation of ChiWIS; L.S. led the design of ChiWIS; K.D.L., B.W.C., and L.S. built and operated ChiWIS; K.D.L., B.W.C., L.S., and A.N. analyzed raw ChiWIS data to produce water vapor measurements; H.S. provided and operated multipass optics; H.S. and O.M. operated AIDA during the IsoCloud campaign; H.S.
5  and O.M. provided and interpreted AIDA instrument data; V.E. provided SP-APicT and APeT data; and B.W.C. and E.J.M. wrote the paper.

*Competing interests.* The authors declare that they have no conflict of interest.

*Acknowledgements.* The authors acknowledge the many individuals who contributed to the IsoCloud project, including Stephanie Aho, Naruki Hiranuma, Erik Kerstel, Benjamin Kühnreich, Janek Landsberg, Eric Stutz, and Steven Wagner, as well as the AIDA technical staff and support team who made this work possible. Eric Jensen, Martina Krämer, and Andrew Gettelman provided helpful discussions and
10  comments. This work was supported by the National Science Foundation (NSF) and the Deutsche Forschungsgemeinschaft (DFG) through the International Collaboration in Chemistry program (NSF grant #CHEM1026830 and DFG grants MO 668/3-1 and EB 235/4-1) and by the NSF through the Partnerships in International Research and Education program (grant #OISE-1743753). K.D.L. acknowledges support from a National Defense Science and Engineering Graduate Fellowship and an NSF Graduate Research Fellowship and L.S. acknowledges support from a Camille and Henry Dreyfus Postdoctoral Fellowship in Environmental Chemistry.



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
