# Peer review of "No anomalous supersaturation in ultracold cirrus laboratory experiments"

_Atmospheric Chemistry and Physics, 2019_

## Referee Comment (RC1) · Anonymous Referee #2 · 6 May 2019

**GENERAL REMARKS**

The study presented in the manuscript investigates potential processes which may cause the observed supersaturation with respect to ice in ultracold cirrus clouds. Whereas these supersaturations have been reported from field observations, the presented study uses for the first time the facilities of the AIDA cloud and aerosol chamber to investigate potential processes responsible for creating these supersaturations. The key research question is whether or not metastable ice whose vapour pressure exceeds that of hexagonal ice, may dominate the formation of ultracold cirrus.

The experiments are very carefully designed, and the uncertainties of applied instruments and modelling tools are assessed in-depth. Overall, the study is scientifically sound and makes a significant contribution to the research field of high-altitude cirrus

clouds. The manuscript is very well organized and fits perfectly to the scope of the journal. Before publication, only minor revisions should be considered which are listed below.

SPECIFIC COMMENTS

As mentioned in the introduction section, several studies suggest as possible explanation for observed ice-supersaturations in cirrus clouds, that uptake rates on ice crystals for typical conditions in cirrus clouds (low temperatures, low ice crystal number concentrations, small ice crystals) are slow and cause long times for relaxation to saturation, so that in lifting air masses a dynamical equilibrium is reached. In the manuscript, the term "pseudo-equilibrium" is used. The authors may consider whether or not the term "dynamical equilibrium" is more appropriate, or discuss the issue.

MINOR ISSUES

Sometimes the unit ppm is used for the volume mixing ratio, and sometimes ppmv. I suggest using ppmv (ppbv) throughout the manuscript, including axis labels.

Page 1, line 21: here the reference to MK should be added.

Page 3, line 31: The sentence seems to be incomplete. I guess it should read: ". . . ice should grow exclusively in hexagonal layers, regardless of the nucleation method . . ."

Page 5, Figure 1 and page 7, line 1ff: In Fig. 1, 5 thermocouples are shown but only the use of four is explained. For completeness, it may be worth mentioning the use of thermocouple T5.

Page 6, line 30: uncertainty from Poisson statistics is proportional to 1/sqrt(n). That should be written here.

Page 11, line 9: the term "particle #" is not explained and may be replaced by "particle number".

Page 15, line 6: the reference to Schilling et al. (2006) should be given here, not only

the name of the first author.

---

## Referee Comment (RC2) · Anonymous Referee #1 · 25 Jun 2019

First of all, my apologies for my late review and delaying the process.

This paper describes an experimental study focused on addressing the hypothesis that a metastable form of ice might form in cold cirrus clouds and that this ice has a larger vapour pressure. The paper is well written and well presented. Overall, the conclusions are well supported and the caveats are discussed. I recommend that this paper is published once the following points are addressed.

Abstract: 'If metastable ice forms in ultracold cirrus clouds, it appears to lack the defects and interfaces that are assumed to produce differences in vapor pressure from hexagonal ice.' I agree with the sentiment of this statement, but the vapour pressure of metastable ice is not necesserily defined by defects and interfaces. The vapour pressure is defined by the material's chemical potential, which is related to crystal structure

and in the case of ice, the degree of disorder as well as defects. Ice I is a material with two end members – hexagonal ice and in theory at least cubic ice. A more correct statement would be 'If metastable ice forms in ultracold cirrus clouds, it appears to have a vapour pressure indistinguishable from hexagonal ice to within xx%.' I think the quantification is important. The uncertainty is around 5%, but the authors stress that a vapour pressure of 20% larger would have a substantial impact on water transport into the stratosphere. The measurements constrain this value, but it could be 5% given the uncertainty which is perhaps not insignificant. Using words like 'negligible', as is done in the conclusions, is inappropriate.

Nomenclature: Ih should be ice Ih (i.e. ice one h; and similarly for ice Isd and ice Ic). I realise that this paper is only referring to ice, but it I think it is better to use the correct nomenclature.

Top of page 4. Murphy [2003] should be cited in the discussion of the transformation timescales.

P4, ln 5-8. This discussion on shape needs to be revised. I don't think (Murray and Bertram, 2006) and (Lowe and MacKenzie, 2008) say anything to support the statement made. Also, Furukawa (1982) doesn't suggest crystals with three fold symmetry will form. Furukawa suggest that cubic ice may form octahedral crystals of cubic ice and the subsequent growth may be off the faces of the octahedral. These early references do not refer to stacking disorder and therefore could not have identified the space group for ice Isd. Some older literature does refer to cubic ice and there is threefold symmetry in cubic ice (look at a cube from a corner), but this is an example of getting the right answer for the wrong reasons. I think the first mention of the trigonal symmetry of stacking disordered ice was in Hansen et al. [2008] (first para of section 6). Also, in reference to Murray et al. (2015), they show that large proportion of small crystals had trigonal symmetry in the TTL in the one campaign (or one of very few) where they were sampled [Heymsfield, 1986]. Imaging probes tend to only have the resolution to look at larger particles, hence there are only very few measurements of

small ice crystal shape in TTL conditions.

P2, ln 23. Bogdan has also proposed some mechanisms for the elevated S in TTL cirrus e.g.[Bogdan and Molina, 2010].

Results and fig 5. I think there is an error in Fig 5. S for stacking disordered ice should be T dependent. Murphy and Koop discuss this: 'Cubic ice can be estimated to have a vapour pressure 3 to 11% higher than hexagonal ice at 200 K (Fig. 5). The vapour pressure ratio is just $\exp(\Delta G/RT)$, where $\Delta G = \Delta H - T\Delta S$ is the Gibbs energy difference between hexagonal and cubic ice, and $\Delta H$ and $\Delta S$ are the corresponding enthalpy and entropy differences. Calculations by Tanaka (1998) indicate that the entropy is nearly identical for cubic and hexagonal ice. If so, then the Gibbs energy difference is equal to the latent heat of transformation between cubic and hexagonal ice, and that is what is shown in Fig. 5. The wide variation in measured $\Delta H$ can be attributed to its small magnitude as well as the difficulty in preparing samples of cubic ice that are not contaminated with either amorphous or hexagonal ice.'

Results: Some discussion of what the Shilling et al. value for S means and some context is required. Metastable ice I will not have a single value. It is a disordered state and its vapour pressure will depend on the way the sample was made. Shilling made their ice by first making amorphous ice and then annealing, which was a pragmatic way of making it in a reproducible manner. Ice made through the direct deposition of ice from vapour may well have a different degree of disorder and therefore a different vapour pressure. Murphy reviewed the available data and came up with a range of possible values, these could be shown in Figure 5.

P18, ln 19. 'its effect on vapor pressure and on transfer of water to the stratosphere would be negligible.'. Be quantitative. The effect of the presence of stacking disordered ice is less than 5% (I've estimated this based on the error bars in fig 5).

Conclusions, 1st line: Amend to reflect the uncertainty in the experiments: 'We see no evidence of an anomalous supersaturation in ultracold cirrus formation, of greater than

[Figure]

∼5%, that field measurements had suggested.

Minor corrections:

P15, ln 6. Shilling should be Shilling et al. (xxx).

P17. Ln 1. 'depresses implied', insert 'the'.

P2 ln 24. Add Murray [2008] to the refs stating that aerosol in a glassy state might inhibit ice nucleation.

P2 ln 29. Murray et al. [2010] also showed how small ice concentrations resulting from heterogeneous nucleation can yield elevated S in clouds.

Citations

Bogdan, A., and M. J. Molina, The Journal of Physical Chemistry A, 114, 2821-2829, 2010.

Hansen, T. C., et al., J. Phys.: Condens. Matter, 20, 285104, 2008.

Heymsfield, A. J., J. Atmos. Sci., 43, 851-855, 1986.

Murphy, D. M., Geophys. Res. Lett., 30, 2003.

Murray, B. J., Atmos. Chem. Phys., 8, 5423-5433, 2008.

Murray, B. J., et al., Nature Geosci., 3, 233-237, 2010.

---

## Author Comment (AC1) · 25 Nov 2019

We thank both referees for their useful comments, and apologize for the long delay in replying. We took the opportunity to refit our data using the new HITRAN 2016 parameters for water, which took more time than expected. Our results are unchanged due to this refit, but the uncertainty is somewhat smaller. We have also included a more thorough treatment of the uncertainty. Responses to detailed comments are listed below. The reviewer's comments are listed in italics and our response in plain text.

**Replies to Referee #1**

In general, in response to reviewer comments we have used more quantitative language in describing results, and have adjusted nomenclature throughout.

Abstract

*A more correct statement would be 'if metastable ice forms in ultracold cirrus clouds, it appears to have a vapour pressure indistinguishable from hexagonal ice to within xx%'*

We have adopted the language suggested by the referee and included a quantitative statement that our measurements constrain anomalous supersaturation to within 4.5%.

Top of page 4. *Murphy [2003] should be cited in the discussion of the transformation timescales*

We now cite Murphy (2003) in the discussion of the transformation timescales.

P4, ln 5-8

We have completely revised this section of the introduction and thank the referee for calling attention to it.

*I don't think (Murray and Bertram, 2006) and (Lowe and MacKenzie, 2008) say anything to support the statement made.* — We have deleted this very confusing sentence. The citations of Murray and Bertram and Lowe and Mackenzie were

originally in place to indicate that the authors had either suggested or observed regions in Earth's atmosphere that were potentially cold enough to support the nucleation of metastable ice, but as written we understand that it gave the wrong impression.

*Also Furukawa (1982) doesn't suggest crystals with threefold symmetry will form. Furukawa suggests that cubic ice may form octahedral crystals of cubic ice and the subsequent growth may be off the faces of the octahedral... ...example of getting the right answer for the wrong reasons.* — Thank you for this clear and concise explanation of the issues with this passage. We have removed the reference to Furukawa, as this is not really the appropriate place for a discussion of cubic ice. (We do still reference his work in the conclusion to note that the form of ice when it first nucleates may influence subsequent growth.)

*I think the first mention of the trigonal symmetry of stacking disordered ice was in Hansen et al. [2008].* — Thank you for pointing out this important detail, I had missed it when reading this paper. Hansen is now appropriately cited for his work.

*Also, in reference to Murray et al. (2015)... ...small ice crystal shape in TTL conditions.* — Included now is a brief discussion of observations of ice crystals with threefold symmetry, and we note Murray (2015)'s conclusion that these crystals are consistent with trigonal habit. We also note that some observations (such as Heymsfield 1986) have shown these crystals to be quite common.

P2, ln 23. *Bogdan has also proposed some mechanisms for the elevated S in TTL cirrus e.g. [Bogdan and Molina, 2010].* — We now cite Bogdan and Molina (2010) and their work on multi-component aerosols.

Results and fig 5 *I think there is an error in Fig 5. S for stacking disordered ices should be T dependent. Murphy and Koop discuss this...*

The referee is correct that S should be temperature dependent. We have now included a temperature dependence. To estimate it, we made the assumption that there is no entropy difference between ice Ih and ice Isd (based on Tanaka's calculations showing minimal entropy different between Ih and Ic), and that we can therefore calculate $\Delta G$ from Shilling's work, and extrapolate that to higher temperatures. This is discussed in the text in the first paragraph of the Results section. We also reference the 3-11% enthalpy differences calculated for cubic ice in Murphy-Koop (2005) in the following paragraph.

Results *Some discussion of what the Shilling et al. value for S means and some context is required.*

We have added context to the first paragraph of the Results section to make it clear to the reader that metastable ice likely does not have a single vapor pressure value, but probably has a range of values that are likely heavily influenced by sample preparation.

P18, ln 19 *'its effect on vapor pressure and an transfer of water to the stratosphere would be negligible.'. Be quantitative. The effect of the presence of stacking disordered ice is less than 5% (I've estimated this based on the error bars in fig 5).*

We have revised to include a quantitative statement of our limits of the effect on vapor pressure of stacking disordered ice.

Conclusions, 1st line: *Amend to reflect the uncertainty in the experiments. 'We see no*

*evidence of an anomalous supersaturation in ultracold cirrus formation of greater than 5%.*

We have included a quantitative statement reflecting the uncertainty in the experiments in the first line of the conclusions.

Minor Corrections

*P15, ln 6. Shilling should be Shilling et al. (xxx)* — We have substituted in the correct citation.

*P17. Ln 1. 'depresses implied', insert 'the'.* — Done.

*P2 ln 24. Add Murray [2008] to the refs stating that aerosol in a glassy state might inhibit ice nucleation* — We have added this citation into the relevant statement.

*P2 ln 29. Murray et al. [2010] also showed how small ice concentrations resulting from heterogeneous nucleation can yield elevated S in clouds.* — We have incorporated this citation into the relevant statement on elevated S in clouds.

**Replies to Referee #2**

Specific Comment *As mentioned in the introduction section, several studies suggest as possible explanation for observed ice-supersaturations in cirrus clouds, that uptake rates on ice crystals for typical conditions in cirrus clouds (low temperatures, low ice crystal number concentrations, small ice crystals) are slow and cause long times for*

*relaxation to saturation, so that in lifting air masses a dynamical equilibrium is reached. In the manuscript, the term "pseudo-equilibrium" is used. The authors may consider whether or not the term "dynamical equilibrium" is more appropriate, or discuss the issue.*

Thanks for calling attention to this issue. Dynamical equilibrium is more common and easily understood terminology, and we have adopted it throughout the text.

Minor Issues

*Sometimes the unit ppm is used for the volume mixing ratio, and sometimes ppmv. I suggest using ppmv (ppbv) throughout the manuscript, including axis labels.* — We have revised to use ppmv and ppbv throughout, including in figures.

Page 1, line 21: *here the reference to MK should be added.* — We have added the reference to MK to this line.

Page 3, line 31: *The sentence seems to be incomplete. I guess it should read: "... ice should grow exclusively in hexagonal layers, regardless of the nucleation method..."* – We have corrected this error in the text.

Page 5, Figure 1 and page 7, line 1ff: In Fig. 1, *5 thermocouples are shown but only the use of four is explained. For completeness, it may be worth mentioning the use of thermocouple T5.* — The topmost thermocouple (T5) measures a known warm spot in the chamber, which is unrepresentative of the levels at which the experiments take place. We have made this clear in the main text and the figure caption describing the

thermocouples.

Page 6, line 30: *uncertainty from Poisson statistics is proportional to 1/sqrt(n). That should be written here.* — We have clearly stated now in the text that these should follow Poisson statistics and that the uncertainty is proportional to 1/sqrt(n)

Page 11, line 9: *the term "particle #" is not explained and may be replaced by "particle number".* — We have replaced '#' with 'number'

Page 15, line 6: *the reference to Schilling et al. (2006) should be given here, not only the name of the first author.* — We have correctly inserted the citation here.